# The Clinical Significance of Transfer RNAs Present in Extracellular Vesicles

**DOI:** 10.3390/ijms23073692

**Published:** 2022-03-28

**Authors:** Daniel S. K. Liu, Qi Zhi Clayton Yang, Mohammad Asim, Jonathan Krell, Adam E. Frampton

**Affiliations:** 1Division of Cancer, Department of Surgery & Cancer, Imperial College London, Hammersmith Hospital Campus, Du Cane Road, London W12 0HS, UK; daniel.liu08@imperial.ac.uk (D.S.K.L.); qi.yang17@imperial.ac.uk (Q.Z.C.Y.); j.krell@imperial.ac.uk (J.K.); 2Department of Clinical and Experimental Medicine, Faculty of Health and Medical Sciences, University of Surrey, Guildford, Surrey GU2 7WG, UK; m.asim@surrey.ac.uk; 3HPB Surgical Unit, Royal Surrey County Hospital NHS Foundation Trust, Guildford, Surrey GU2 7XX, UK

**Keywords:** extracellular vesicle, exosome, transfer RNA, tRNA fragment, tRNA half, cancer

## Abstract

Extracellular vesicles (EVs) are important for intercellular signalling in multi-cellular organisms. However, the role of mature transfer RNAs (tRNAs) and tRNA fragments in EVs has yet to be characterised. This systematic review aimed to identify up-to-date literature on tRNAs present within human EVs and explores their potential clinical significance in health and disease. A comprehensive and systematic literature search was performed, and the study was conducted in accordance with PRISMA guidelines. Electronic databases MEDLINE and EMBASE were searched up until 1 January 2022. From 685 papers, 60 studies were identified for analysis. The majority of papers reviewed focussed on the role of EV tRNAs in cancers (31.7%), with numerous other conditions represented. Blood and cell lines were the most common EV sources, representing 85.9% of protocols used. EV isolation methods included most known methods, precipitation being the most common (49.3%). The proportion of EV tRNAs was highly variable, ranging between 0.04% to >95% depending on tissue source. EV tRNAs are present in a multitude of sources and show promise as disease markers in breast cancer, gastrointestinal cancers, and other diseases. EV tRNA research is an emerging field, with increasing numbers of papers highlighting novel methodologies for tRNA and tRNA fragment discovery.

## 1. Introduction

Extracellular vesicles (EVs) are small cargo-containing structures with a lipid bilayer but do not have the cellular machinery required to replicate. They have been shown to play a role in cell-to-cell communication, as they can be found to transport biological material including proteins, lipids, ribonucleic acid (RNA), and deoxyribonucleic acid (DNA) between cells, leading to cellular changes within multi-cellular organisms [1]. It is a process that has been conserved through evolution, found both in prokaryotes and eukaryotes [2].

EVs are an umbrella term for a heterogenous array of secreted membrane vesicles and can be further distinguished by cell source, biogenesis pathways, and size ranges, thus contributing to diverse nomenclature [2]. Two broad categories that are often used distinguish microvesicles (MVs) and exosomes on the basis of cellular biogenesis [2]. Microvesicles are generated by the outward budding of the phospholipid bilayer generating vesicles made directly from the plasma membrane. These vesicles tend to be larger with MVs having a size range of up to 1000 nm or larger in some cases [2,3]. Exosomes, on the other hand, involve intracellular mechanisms forming multivesicular endosomes (MVEs) which contain intraluminal vesicles (ILVs). ILVs are formed from the budding of MVEs which then fuse with the plasma membrane and get released extracellularly as exosomes. Exosomes are usually smaller in size (<150 nm) closely reflecting that of ILVs [2,3] and therefore also express specific markers that relate to their endosomal origin [2].

EVs have been found to be present in virtually all types of biofluids, including cerebrospinal fluid (CSF) [4], urine [5], blood [6], bile [7], and breast milk [8] as well as being released by other organisms and even plants [9]. Numerous studies focus on their carrier material which contains lipids, protein, and RNA in an endeavour to link them to the pathophysiology of a wide range of diseases [1]. With the current advances in RNA sequencing technologies and bioinformatic strategies, this has drawn attention to the wide range of small RNA (sRNA) content in EVs. 

Whilst much of the literature has focused on studying micro RNAs (miRNAs) [10], short 16–22 nucleotide (nt) single-stranded molecules with a role in canonical transcription and translational pathways, less is known about the role of transfer RNAs (tRNA) despite being the most abundant RNA in the human genome [11], and the first non-coding RNA to have been discovered [12]. The canonical cytoplasmic tRNA molecule is 76 nt long and well-conserved, with a cloverleaf secondary structure and a tertiary L-shaped structure that separates the anticodon triplet for recognition of template mRNA from the amino acid attachment site. Whilst tRNAs are present in all species for protein translation, tRNA genetic expression is species-specific and with over 270 different tRNA sequences present among approximately 450 tRNA human genes, highly complex in eukaryotes [13]. Furthermore, the genetic expression of tRNAs is subject to modifications such as base-specific methylation, altering function, and fragmentation which can generate whole new subspecies of RNA molecules. This has led to, over the past decade [14,15], a wide expansion in tRNA biology with regulatory functions affecting gene expression, protein synthesis, and the stress response as significant downstream functions [11].

The fragmentation products of tRNAs are a potential source of new functionality away from conventional protein synthesis. tRNA-halves and tRNA-derived fragments (tRFs) are two such fragmentation products of tRNAs and have distinct biogenesis pathways (Figure 1). tRNA-halves are thought to be formed by nucleases such as angiogenin as well as Dicer and RNase Z which are upregulated under varying cellular conditions including stress [11,14,16]. The simplest fragmentation is a cleavage of the anti-codon loop forming either 5′ or 3′ tRNA-halves with lengths corresponding to half the full-length of mature tRNAs [14]. tRFs on the other hand, includes 5′ and 3′-tRFs with shorter nucleotide sequences and are named based on where the cleavage occurred, and are thought to be produced from the cleavage occurring in or around the D-loop or T-loop structure of mature tRNAs, respectively [16]. However, tRFs can also arise from 3′ nuclease cleavage of 5′ tRNA-halves and as such, the enzymes responsible for the production of smaller tRFs remain not well understood [16]. There are also tRFs that do not correspond to the 5′ or 3′ end and are called internal tRFs. Downstream effects of tRFs are thought to be related to RNA machinery, associating with argonaute proteins to exhibit miRNA-like regulatory and silencing activity on mRNA to facilitate post-transcriptional repression [16]. This represents a new spectrum of molecular pathways that converge to modify gene expression and thus influence cells in health and disease.

In this review, we aimed to comprehensively search the literature for studies that have identified tRNA and tRNA-derived cargo present in human EVs from studies performed either in vivo or in vitro. This has enabled us to clearly define the current state of EV tRNA research, defining the EVs present in these studies and highlighting functional and clinical significance where noted. 

## 2. Results

A total of 60 studies were identified as relevant for this review. Forty-four studies (73.3%) focussed on pathological diseases, whilst the remaining 16 studies (26.7%) did not investigate a particular disease but instead focussed investigations on optimising methodologies such as choice of EV source, isolation method, characterisation of tRNA cargo, as well as on biofluids from healthy individuals. 

Of the 44 disease-focussed studies, 43.2% of them were primarily investigating cancers (Figure 2A). The remaining studies are mainly comprised of papers focussing on respiratory, renal, bone, sepsis, autoimmune, reproductive tract, and infectious diseases. Notably, gastrointestinal cancers were the most common cancer studied (31.6%), followed by breast cancer, pan-cancer, and HPV-induced cancers representing 15.8% each (Figure 2A).

### 2.1. Overview of EV Isolation Methodologies and Characterisation

A total of 71 different EV isolation protocols were identified from 60 studies (Figure 2B). The main methodologies used were ultracentrifugation (UC), precipitation, affinity-based purification, size exclusion chromatography (SEC), density gradient, and sequential filtration, with the majority using either UC or precipitation in isolation (77.5%). In order to simplify these protocols for comparison, we excluded all centrifugation steps lower than 16,500 g and did not include large pore filtration as a distinct step. The extracted protocols are shown in Appendix A.

Precipitation was the most popular EV isolation method identified, with 46.5% of protocols using it as the sole EV isolation method (Figure 2B). Six different precipitation methods were used with ExoQuick (System Biosciences, Palo Alto, CA, USA) being the most common kit used. Other kits included Total Exosome Isolation Reagent (Invitrogen, Waltham, MA, USA), PureExo Exosome Isolation Kit (101Bio, Mountain View, CA, USA), miRCURY Exosome Isolation Kit (QIAGEN, Hilden, Germany), and Ribo Exosome Isolation Reagent Kit (Ribobio, Guangzhou, China) and Plasma/Serum Exosome Purification Kit (Norgen Biotek Corporation, Thorold, ON, Canada), Cell Culture Media Exosome Purification Mini Kit (Norgen Biotek Corporation), Umibio Exosome Isolation Kit (Umibio, Shanghai, China) and Exo2D RNA solution (Exosomeplus, Seoul, Korea). 

The second most common methodology to isolate EVs was UC (31%) as a sole method, with affinity purification methods and SEC, third and fourth, respectively (Figure 2B). Protocols for UC varied and there was minimal reporting of rotor sizes and centrifuges used which has an effect on pellet aggregation and EV isolation. Most (90.5%) used a final spin speed of >100,000 g and of the 22 UC protocols (used as the sole method), 36.4% of protocols involved two consecutive UC spins. In addition, seven protocols used a hybrid of methods with the most common (four of them) being the combination of density gradients and ultracentrifugation (Figure 2B).

Minimal information guidelines for the reporting of EV studies were published in 2014 and updated in 2018 which highlighted characterisation criteria needed to demonstrate the presence of EVs in a study. We therefore reviewed the studies for Western blots showing EV markers, transmission electron microscopy pictures, and evidence of nanoparticle tracking analysis (or equivalent quantitative analysis). In total, 35% of the studies identified employed these three EV characterisation techniques. 

### 2.2. Overview of EV Sources

The vast proportion of studies identified used either cell lines or blood as their source of EVs, with the majority still obtained from cell-conditioned media. A total of 40.8% of protocols used cell lines, and 45.1% of protocols used blood (Figure 2C). Other sources were surprisingly diverse, coming from semen, saliva, ovarian follicular fluid, bile, sweat, and urine.

### 2.3. EV tRNAs Present in Non-Pathological Samples 

#### 2.3.1. Optimisation of EV Experimental Methodology

Several studies are now discussed which had the broad remit of optimising various stages of the pipeline required for EV tRNA research. This includes determining the parameters of EV source material, tRNA isolation, and RNA sequencing methodologies from primarily healthy samples.

Srinivasan et al. [17] investigated comprehensively the RNA profiles of different biofluids in a large study that involved 20 healthy volunteer donors and 3 patients with cholangiocarcinoma (CCA). They undertook RNA sequencing on five different biofluids (bile, cell culture, plasma, serum, and urine) and applied ten different extracellular RNA isolation methods with the compatibility of each protocol matched to the type of biofluid investigated. Additionally, this study was unique in being performed across six different laboratories to determine inter-laboratory variability. tRNA expression profiles were grouped according to the type of fragment (5′-half, 3′-half, 5′-tRF, 3′-tRF, or internal, i-tRFs) and they noticed significant differences between samples, with bile and urine samples almost completely comprised of 5′-halves and 5′-tRFs, whilst cell-line-derived EV tRNA sequences were predominantly made up of 5′-tRFs, compared to their intracellular profile which contained mainly 3′-tRFs and i-tRFs. According to amino acid distribution, each biofluid type had broadly similar profiles, with 30–40% being represented by both tRNA-Gly and tRNA-Glu and 2–5% by tRNA-His and tRNA-Val. By comparing isolation methods, they identified that across EV samples, these were less variable in profile compared with the analysis undertaken for miRNA and mRNAs.

This variability in EV isolation methods was also examined by Lasser et al. [18], who looked at just one type of EV released by mast cells using UC and subsequent sucrose density gradient to isolate high density (HD) and low density (LD) EVs. The density gradient is a highly selective, less commonly used technique for EV isolation. They found that in the HD fraction, mature miRNA (23%) was mainly found, whereas the LD fraction showed a predominance of tRNAs (28%). Moreover, 96% of the tRFs found in the LD fraction were also mainly from the 5′ terminal of tRNAs. This is significant as UC inherently takes into account particle and vesicle density, which means that different UC protocols or even rotors may contribute to an alteration in the size and density of vesicles isolated, leading to variability in RNA profiles. Further analysis of centrifugation techniques was shown by Tosar et al. [19] who analysed breast epithelial EVs (from MCF-7 and MCF-10A cells) dividing them into a 16,000 g pellet (p16) and 100,000 g pellet (p100), and comparing these to the supernatant (S100). The authors found that the S100 portion contained an 18-fold increase in tRNA fragments compared to the intracellular fraction, with the p16 and p100 fractions having a 3-fold and 2-fold increase, respectively. These extracellular tRNA-derived sequences showed a narrow size distribution, with a mode at 30–31 nt and again a preference for 5′ sequences. In contrast, miRNAs decreased in relative abundance between the intracellular and extracellular (vesicular and non-vesicular) compartments. What was interesting was that they noted that extracellular tRNAs may be present and bound to ribonucleoprotein-associated complexes which complicates evaluation of the EV compartment. Additionally, they identified that 5′ halves from tRNA-Glu-CUC were detectable in foetal bovine serum, a common supplement to cell culture media and so care must be taken in excluding this from EVs isolated from cell lines.

Sork et al. [20] investigated five immortalized cell lines (HEK293T (human embryonic kidney), RD4 (human skeletal muscle cells), Neuro2a (neuroblastoma cells), C17.2 (immortalised mouse neural progenitor cells), and C2C12 (immortalised mouse myoblasts)) to gain a better comprehensive understanding of the differences and overlap in RNA species and abundance in EVs, using transcriptomic and proteomic approaches. They found that tRNAs were one of the most predominant RNAs, after ribosomal RNA (rRNA) and sRNA, found in the EVs across all five cell lines. Specifically, tRNA-Gly-GCC, tRNA-His-GTG, and tRNA-Glu-CTC were the most prevalent fragments. The authors were, however, not able to further subclassify these tRNA fragments and were also not able to attribute any to specific biological functions.

Small RNA library preparation was investigated by Huang et al. [21] in 2013, as this can be a source of RNA sequencing variability. They compared the applicability of three RNA sequencing methods within the plasma EVs of healthy volunteers obtained using ExoQuick. Three preparation kits designed to obtain small non-coding RNAs (30–40 nts) were investigated (NEBNext multiplex small RNA library preparation kit, NEXTflex small RNA sequencing kit, and TruSeq small RNA sample preparation kit) and appeared to show good technical reproducibility. However, each kit had preferential enrichment of low abundant RNAs. Importantly, this inherent bias in RNA library preparation would be present for the tRNA analysis performed in their study as tRNAs accounted for just 1.24% of total mapped reads. It is highly recommended that all studies investigating EV RNAs should therefore include validation by qRT-PCR.

Since then, a couple of new techniques for analysis of EV tRNAs have arisen, with a new pipeline—One-Step Total Transcriptome Protocol proposed by Amorim et al. [22] to address the issue of analysing both short and long RNAs found in EVs in a single next-generation sequencing (NGS) library. This modified protocol used RNA fragmentation by RNase-III treatment to minimise adaptor-dimer formation and changes in the ligation and PCR step. The authors found that within the plasma EVs of five healthy participants included in the study, short non-coding RNA represented the second most abundant RNA population after rRNA reads were filtered. Of note, tRNAs were the major species, representing 57.29% of mapped reads. This pipeline reduced bias from amplification due to heterogenous samples and therefore has the potential for applications in liquid biopsies, as it allowed for accurate total transcriptome profiling of diverse long and short RNAs. Shurtleff et al. [23], used a novel thermostable group II intron reverse transcriptase (TGIRT) sequencing method to investigate the RNA profiles of EVs secreted from human embryonic kidney 293T cells in order to identify the role of YBX1 RNA-binding protein in sorting small non-coding RNAs into EVs which they had previously identified as being necessary for the sorting of miR-223 in the same cells [24]. This method allowed RNA species with post-transcriptionally modified secondary structures to be better reverse transcribed for library preparation. EVs were isolated using a highly selective hybrid method, using UC and sucrose gradient, and appeared to contain mainly full-length mature tRNA transcripts representing 50% of the total RNA reads. Surprisingly, EV miRNAs constituted <0.5% of total reads using this methodology. Additional work to check for 3′ phosphates showed that tRNA fragments were not abundant in these EVs. Their paper also made note of evidence for an exosome-specific tRNA modification involving a tRNA modification at the D-loop (see Figure 1). Finally, to characterise the function of YBX1 in EVs, they conducted sequencing on wild-type cells and YBX1-null cells, finding a 69% decrease in tRNAs in YBX1-null exosomes and a corresponding 2.3-fold increase in tRNAs in YBX1-null cells, suggesting that YBX1 is a key protein in the sorting of tRNAs into EVs in HEK293T cells.

Therefore, recent focus has now been on how selectively tRNAs are sorted into EVs. A study from 2020 by Gambaro et al. [25] used YBX1-null MCF-7 cells and appeared to contradict the findings of Shurtleff et al. [23] with regard to the importance of YBX1. Synthetic oligonucleotide 5′-tiRNA-Gly was used to show that tRNA-halves could be secreted from the parent cell into EVs in a concentration-dependent manner without the need of specific sequence motifs. 5′-tiRNA-Gly are capable of forming RNase-resistant homodimers and this was previously demonstrated to be important to the process. They were able to also demonstrate cell-to-cell transfection of synthetic 5′-tiRNA-Gly transported in EVs which led to almost two hundred gene alterations seen by the microarray panel. This provides the exciting groundwork for EV-mediated therapeutics to incorporate synthetic oligonucleotides. Kaur et al. [26] investigated EV surface markers as another potential mechanism of RNA sorting to EVs through differences in RNA populations present in EVs with CD63, CD47, and/or MHC1 surface markers and secreted by Jurkat T cells. The authors found that MHC1+ EVs compared to CD63+ and CD47+ EVs, showed a distinct RNA profile with an increased abundance of tRNAs expressed. tRNA-Glu-CTC 1-5 and tRNA-Glu-TTC 3-1 (reported as TRE-CTC and TRE-TRR) were the full-length tRNAs that were specifically present in higher levels in MHC1+ EVs. The majority of EVs captured in the study, however, did not have any of the aforementioned surface proteins, which, therefore, limited the analysis from clearly identifying the role of CD63, CD47, and MHC1 in EV tRNA loading. Nozaki et al. [27] postulated that the nuclear protein Poly ADP-ribose polymerase (PARP) 1 may also have a role in intercellular signalling using EVs as well as its known roles in tumour proliferation, DNA repair, and inflammation. They investigated EVs produced from embryonic stem cells (ES) and found EVs from PARP1 knockout cells had higher levels of tRNAs than EVs from wild-type PARP1 cells (49.1% and 39.5% of the total sRNAs, respectively). No further functional analysis was demonstrated.

#### 2.3.2. EV tRNAs Are Present in Diverse Biofluids

Ferrero et al. [28] analysed the RNA profiles across four different biofluids (blood plasma, urine, cervical scrapes, and stool) from healthy individuals to determine if the RNA profile remained consistent across different samples. However, blood plasma was the only biofluid where EVs were isolated, whilst the other biofluids were subject to RNA isolation methods directly. Here, tRNAs were found to make up a high proportion of the circulating RNA species present in urine (45.1%) compared with other specimens (1–3.3%) including plasma EVs. A study by Miranda et al. [29], published in 2014 also analysed the RNA repertoire present within the microvesicles of urine of an individual healthy male isolated by UC. They found that a large proportion of reads was ascribed to ribosomal RNA (87%), which in some studies is normally filtered, and although a wide variety of species were present including tRNA, snRNAs, small cytoplasmic RNAs, satellite repeats, and others, this comprised less than 6–6.1% of the total number of aligned reads in the sample. 

We identified three studies that analysed EV tRNAs in unique biofluids. Bart et al. [30] characterised for the first time the nucleic acid profile associated with EVs from sweat produced under rigorous exercise. EV-associated-tRNAs were measured individually and pooled, finding that tRNAs were found to be the most abundant RNA species found (accounting for over 70% of annotated reads) and corroborated by individual sweat EV analysis. Vojtech et al. [31] was the only study identified characterising the non-coding RNA profile of EVs from semen in order to investigate their potential immunomodulatory functions. Six samples were isolated using UC with a sucrose cushion and sequenced as two libraries based on size (15–40 nt and 40–100 nt) fractionated using a 10% acrylamide-urea gel. Full-length mature tRNAs accounted for 5.93% of the reads from the larger size library whilst tRNA fragments accounted for 16.02% of the small size library. Different tRNA fragment isoacceptor types were also found selectively enriched as full-length or fragments, with tRNA-Glu and tRNA-Asp showing the biggest difference between the two in proportion. Only nine different 5′-tRNA fragments were tested, but 5′-tRNA-Gly and tRNA-Val predominated. Most tRNA fragments from both libraries were again noted as originating from the 5′-tRNA ends. These profiles appeared to be fairly consistent in the small sample that was tested. Additionally, a study by Cooke et al. [32] characterised the tRNA profile in the EVs isolated from the syncytiotrophoblast (STB) of the placenta. Six samples of placental tissue were collected during elective caesarean section and perfusate was obtained through a collecting system. STB EVs were compared to placental tissue and showed tRNAs representing over 95% of mappable reads, in contrast to just 50% of the reads at a cellular level. 5′-tRNA halves were found to be the predominant tRNA species found in STB EVs, which was validated by qRT-PCR.

One notable final study by Zhao et al. [33] investigated the differences and similarities in the serum exosomes between pooled samples from 4–5 humans, mice, and rats. They found tRNAs and miRNAs were the most abundant RNAs across the three species investigated. 

### 2.4. EV tRNAs Implicated in Disease

#### 2.4.1. Enrichment and Differential Expression of EV tRNAs in Cancer

One of the largest areas for EV tRNA research is in cancer diagnosis and pathogenesis with 19 studies identified in our review. EVs are one way by which malignant cells communicate and signal, thus underpinning several of the critical hallmarks of cancer first detailed by Hanahan et al. [34].

One of the earliest and broadest studies to be identified is Yuan et al. [6], who investigated the EV RNA cargo by sequencing the blood plasma of 50 healthy patients and 142 pan-cancer patients, including those with colon, prostate, and pancreatic cancer. This analysis found a very low abundance of tRNAs, representing just 2.1% of total mappable reads, and a high proportion of miRNA and piwi-interacting RNA (piRNA). No mention was made of differentially expressed tRNA markers. The authors did however note that different methods for vesicle isolation, RNA extraction, sequencing library preparation, or gel size selection all contributed to significant batch effects which could contribute to bias in such a large dataset. Similar EV tRNA proportions were noted by a smaller pan-cancer study in 2021 conducted by Jia et al. [35]. They investigated the difference in extracellular RNA in different plasma fractions. Plasma was obtained from 10 patients with a variety of cancers, namely lung cancer, tongue cancer, prostate cancer, and colon cancer. They used ExoQuick precipitation to generate a series of samples together with a final supernatant. The authors found that tRNAs showed a decrease in proportion, accounting for 4.3% of the RNA content in the “EV” precipitate. However, they did not characterise their EVs, which limits the conclusions we can draw from this. Vergauwen et al. [36] used a combination of SEC and density gradient to isolate EVs from blood plasma obtained from a broad range of patients with ovarian cancer, breast cancer, and HIV. This was in an attempt to define clinical context-dependent variations in the EV RNA profile in cancer patients. tRNAs were found to comprise 7.79% of non-ribosomal mapped reads compared with 7.18% for miRNAs. They also investigated the EV RNA profiles of pooled samples across different time points (several months after therapy or surgery) and found that the tRNA species carried in plasma EVs were indeed time-dependent.

The largest study identified in cancer included over 100 breast cancer patients and was published by Wang et al. [37] in 2020. This involved 176 early-stage breast cancer (BC) patients together with 140 female controls and is notable for its primary focus on the tRNA profile (tRFs and tRNA-halves) found from cell supernatants, plasma exosomes, and tissue samples. This study utilised several stages of sequencing followed by quantitative reverse transcription polymerase chain reaction (qRT-PCR) to narrow down potential candidates with differential expression. Six 5′-tRFs were eventually identified as downregulated in BC (tRF-Glu-CTC-003, tRF-Gly-CCC-007, tRF-Gly-CCC-008, tRF-Leu-CAA-003, tRF-Ser-TGA-001, and tRF-Ser-TGA-002) and this was verified in an external validation phase (samples from another hospital). Whilst this was focussed on extracellular plasma markers, a subset of 40 patients was used to examine the exosomal compartment, and two of these markers (tRF-Ser-TGA-001 and tRF-Ser-TGA-002) exhibited significantly lower levels in exosomes. This was a less marked difference than circulating plasma expression which indicated that exosomes may not be the source of these particular tRFs. A supplementary paper [38] by the same authors reported on a similar cohort involving 128 breast cancer patients and 116 healthy controls, this time using BC cells compared with epithelial cells (MCF-10A) to validate upregulated tRF biomarkers identified from sequencing data. This study found three tRNA-derived small RNAs (tsRNAs; tRF-Arg-CCT-017, tRF-Gly-CCC-001, and tiRNA-Phe-GAA-003), which were all validated in a similar exosome subset of 40 patients as well as in circulating plasma samples. Target gene analysis for these tsRNAs showed them to be cancer-related involving TGF-β and Wnt-signalling-pathways as well as regulating genes AKT1, MRAS, and WNT4. The validation of these candidates by qRT-PCR was performed in both studies, strengthening the results found. A similar study by Koi et al. [39] investigated 32 BC patients and 20 controls and found a candidate tRF-Lys (TTT) significantly expressed in serum, but not quite significantly different in EVs (*p* = 0.62). There are a number of reasons for the difference in findings, including the method of EV isolation as well as the threshold for candidate markers. Additionally, there were differences in the proportion of breast cancer subtypes and stages of the included patients in each respective study.

Three studies from China regarding gastric cancer were identified initially. Ren et al. [40] sequenced EVs obtained from gastric cancer cell lines and normal gastric mucosal epithelial cell lines. Their library preparation protocols used were only targeted to reliably capture 20–40 nt sequences; thus, whilst they were able to accurately capture tRFs, the presence of full-length mature tRNAs was not evaluated. This study revealed low read counts of tRFs (<10%) across the five cell lines examined. Tang et al. [41] investigated RNA identified by sequencing from a cohort of 36 early-stage gastric cancer patients and healthy volunteers. They also reported a low tRNA abundance with only 0.04% of total mapped reads, but they were sequencing at low depth, obtaining average mappable reads of about 6.5 million per patient. A larger study on gastrointestinal cancers was conducted recently by Ge et al. [7]. The study comprised 82 patients, investigating the difference in exosomal miRNA between patients with malignant and benign biliary obstruction. The malignant group included pancreatic cancer, cholangiocarcinoma, and ampullary carcinoma patients. Interestingly, although the study focussed on exosomal miRNA, it was actually only the second most mapped sRNA at 24%. The proportion of exosomal sRNA mapped to tRNA was in fact the highest, at 27%.

One final paper we investigated in this area was subsequently retracted by the publishers due to similarities to another article regarding tRNAs [42,43].

Various other studies investigating cancers in the gastrointestinal system were identified. Zhu et al. [44] profiled SK-Hep1 liver cancer cells and showed that 5% of exosomal small RNAs were generated from tRNA, which was higher than the 0.2–2% of total small RNAs found in plasma exosomes from an unreported number of liver cancer patients. In both cases and supporting other identified studies, tRNA-5 (5′-tRF) was also the major type of tRNA with four tRNAs significantly higher in the plasma exosomes of liver cancer patients (tRNA-Val-TAC-3, tRNA-Gly-TCC-5, tRNA-Val-AAC-5, and tRNA-Glu-CTC-5). Kumar et al. [45], published a small study which investigated the profiles of mRNA, piRNA, and tRNAs in the exosomes of blood serum of healthy individuals, intraductal papillary mucinous neoplasms (IPMN), and pancreatic ductal adenocarcinoma (PDAC). Using two donors from each sample, they found five tRNAs differentially expressed between IPMN vs. healthy, PDAC vs. healthy, and PDAC vs. IPMN patients, but these were not validated. Human oesophageal cancer was investigated in a study by Liao et al. [46], reporting the class of sRNAs exosomal tRNA reads represented 9.74% and 32.33% of unique reads and total reads, respectively, although individual tRNAs as potential biomarkers were not mentioned. It was also observed that the number of unique reads and total reads for exosomal miRNAs, 0.63% and 1.31%, respectively, was lower than that of exosomal tRNAs. 

The increased abundance of specific molecules of tRNAs compared to miRNAs was also noted by Wei et al. [47] in glioblastoma stem cells, highlighting that the tRF Glu-CTC was potentially present in EVs at a 1:1 ratio, whilst the most abundant miRNA molecule (miR-21) was present at a lower 1:10 ratio. Contrastingly, lower levels of tRNAs were found compared to miRNA in the study by Li et al. [48], where they investigated the RNA profile of the plasma EVs from 28 patients with lung adenocarcinomas (ADC), 13 lung squamous cell cancer (SQCC), and 13 healthy controls (CTRL). RNA sequencing analysis revealed SQCC, ADC, and CTRL had EV tRNA fragment proportions of 2.95%, 1.13%, and 1.13%, respectively. These results were validated by qRT-PCR using a larger cohort of patients, including some of the patients from the sequencing stage as well as an additional 16 ADC patients, 18 SQCC patients, and 14 CTRL patients. No further investigation into the different tRNA biotypes present within EVs was conducted.

Three studies relating to HPV-induced cancers were also identified. Tong et al. [49] investigated the RNA profile of EVs in head and neck squamous cell carcinoma (HNSCC) and cervical squamous cell carcinoma (CSCC) cell lines. Overall, exosomes and microvesicles were highly enriched with tRNAs when compared to their source cells (31% and 34% vs. 3%), with nearly 90% of the tRNAs made up of 5′-tRNA-halves and 5′-tRNA fragments. These extracellular sources had similar tRNA compositions but differed with respect to their source cells, with broad classifications of tRNA species indicating Glu-CTC being the most abundant species in exosomes and microvesicles (32.6% and 29.9%, respectively) and Gly-GCC in source cells (27.6%). No differences were found, however, across cancer type or HPV status. A large study by Xi et al. [50] in 2021 further investigated the level of expression of extracellular-vesicle-associated tRFs from the blood plasma samples of 30 HNSCC (termed hypopharyngeal in the paper) patients with no metastasis, 30 patients with lung metastasis, and 30 healthy controls. Differential expression analysis and qPCR validation revealed a particular tRF (tRF-1:30-Lys-CTT-1-M2) to be significantly expressed in the EVs of patients with cancer, and this was additionally associated with tumour stage and grade, as well as lung metastatic risk. In addition, Cho et al. [51] investigated the exosomal RNAs from the plasma of 30 cervical cancer patients who were undergoing chemoradiotherapy and compared them to healthy individuals. They found 474 differentially expressed exosomal tRNAs. However, when they used mRNAs to correlate with the relative changes in non-coding RNA, they found that only 9 of the 474 tRNAs were actually functionally relevant. 

EV tRNAs have been identified in different subsets of EVs from malignant melanoma cell lines in two papers by Lunavat et al. [52,53]. Their 2015 paper separated different subsets of EVs namely apoptotic bodies and microvesicles from exosomes using differential centrifugation and analysed their content. They found that exosomes potentially contain relatively more RNAs than apoptotic bodies and microvesicles, but that tRNAs were relatively depleted in exosomes. They specifically found that mitochondrial tRNA (mt-tRNA) was enriched in cells but not in exosomes, but it is unclear if they distinguished this from cytoplasmic tRNA species. In their later study [53], the effects of treatment on malignant melanoma cells harbouring BRAF^V600^ mutations were explored. The authors investigated the effect vemurafenib (BRAF inhibitor) had on the RNA cargo contained within the different subsets of EVs secreted by melanoma cell lines. They once again identified that different subsets of EVs had distinct tRNA enrichment profiles, but this time found that a larger proportion of the reads were mapped as tRNA in the extracellular vesicles compared with cells, with the majority of tRNAs found in microvesicles. Particularly enriched were glutamate and glycine tRNAs, whilst exosomes were enriched with valine tRNAs. Aspartic acid, proline, and alanine tRNAs were depleted in vesicles compared with cells. Interestingly, vemurafenib was not shown to alter the relative expression levels of these tRNAs in the different subsets of EVs. These studies [52,53] therefore noted that treatment did not alter proportions of tRNA found but was not able to shed further light on specific tRNA expression. 

#### 2.4.2. EV tRNAs in Respiratory Disease

Four studies were found in this field by Corsello et al. [54], Sundar et al. [55], and Singh et al. [56] investigating how different smoking methods had a differential impact on the EV tRNA content of cells and in blood samples. Corsello et al. [54] investigated the effect cigarette smoke condensate (CSC) had on the RNA profile of EVs derived from small airway epithelial (SAE) cells. The authors employed an Exospin method which incorporates both precipitation and SEC for EV isolation. Average tRNA proportions increased from 9.1% to 12.5% but this did not reach a statistical difference. They were the largest component of identified RNA species, with miRNAs in second place. Sundar et al. [55] also investigated if patients with chronic obstructive pulmonary disease (COPD), compared with smokers and non-smokers, had a different blood EV tRNA profile. Here, tRNAs were the second most common biotype and differential expression of five tRNAs was found between smokers and COPD, and between non-smokers and COPD. Singh et al. [56] was a further paper by the same research group using the same cohort of samples and went on to conduct a separate comparison based on whether they were non-smokers, cigarette smokers, waterpipe smokers, or e-cigarette smokers. An important point to note however is the lack of detail as to whether these tRNAs were fragments or halves.

#### 2.4.3. EV tRNAs in Sepsis 

Three studies were identified that investigated EV tRNAs in the blood samples of patients in intensive care (ICU) with sepsis and compared with healthy volunteers. The first was a comprehensive study by Buschmann et al. [57] which compared five different kits (four methods of EV isolation namely precipitation, size exclusion chromatography, membrane affinity, and ultracentrifugation) in combination with two RNA extraction methods to isolate EVs from blood plasma. Crucially, they found differences in RNA subtypes by different methods, with more tRNAs captured from EVs using the membrane affinity method (exoRNeasy Serum-Plasma Midi Kit by Qiagen). However, this was noted to be a relatively low specificity kit for EVs. One important conclusion from this paper was that isolation methods less specific for EVs tended to yield more RNAs, better libraries, and an increase in differential expression, compared with more specific methods which purify EVs rather than circulating cell-free material in general, resulting in less complex libraries. The proportion of tRNA yield could therefore be a surrogate measure for specificity (or lack thereof) of EV capture. Another study in this field was by Hermann et al. [58], who found that the majority of the expression profiles of miRNA and tRNAs matched between arterial and venous sera in 20 cardiac surgery patients. They assessed EVs by precipitation (termed crude) and additionally purified these EVs by SEC, thus suggesting that RNAs including tRNAs obtained from either a vein or artery should be suitable for the detection of EV biomarkers. Hermann et al. [59] was able to then conduct a large study including 30 patients with community-acquired pneumonia, 65 with sepsis, and 47 healthy individuals, using precipitation methodology (which had been noted to reliably separate sepsis from healthy controls) to investigate RNA biomarkers associated with EVs. In this paper, tRNAs were found to be less abundant than miRNA species across all three patient groups, with a large proportion of unmapped reads. Unfortunately, they did not further investigate potential tRNA markers in this well-conducted study.

#### 2.4.4. EV tRNAs in the Urine of Patients with Renal Disease

Renal diseases represent an important avenue where EVs could act as diagnostic markers, and EVs isolated from the urine are being evaluated as a non-invasive metabolic snapshot to assess pathological renal cells. We highlight two small in vivo studies in this review investigating urinary EVs from patients with chronic kidney disease (CKD) and idiopathic membranous nephropathy (IMN).

Khurana et al. [5] compared the urinary exosomes obtained by UC from 15 patients with CKD compared with 10 healthy volunteers and identified two tRFs significantly decreased (5′-tRF^Val^ and 5′-tRF^Leu^, displaying a size of about 32–36 nt, i.e., halves). Interestingly, the authors used cell culture and Northern blotting to verify that these tRFs were present in a high abundance in the exosomes released by renal cells. In addition, mitochondrial tRNAs (mt-tRNA; tRNA-Cys, tRNA-Pro, and tRNA-Glu, sized about 60–68 nt) were also found to be significantly decreased in the urinary exosomes but this was not further investigated. Zhang et al. [60] was one of the few research groups to use an immunoaffinity precipitation method to isolate EVs based on affinity to phosphatidylserine (Wako Pure Chemical Industries, Osaka, Japan) and investigated the proportion of different small RNAs found from the urinary exosomes of idiopathic membranous nephropathy (IMN) in a small group of six patients and five controls. They found that small RNAs derived from tRNAs were the most prevalent RNA species found within the urinary EVs of IMN patients. The authors however did not perform a separate experiment to validate the source and origin of the RNA, nor did they identify specific candidates.

#### 2.4.5. EV tRNAs in Metabolic Bone Disease

Degeneration of the bone and osteoblastogenesis are closely linked; thus, perturbations in these biological processes have important implications for bone health. Three EV studies were conducted in this field, where two studies conducted in vivo experiments for the investigation of bone degeneration whilst the other study conducted in vitro experiments on osteoblastogenesis, all reporting changes in the tRNA levels of exosomes as one of their key observations. 

A large study of 80 patients conducted by Zhang et al. [43], using the ExoQuick kit, found potential diagnostic markers for osteoporosis by investigating the differences in exosomal tRNA fragments found in the blood plasma of osteoporosis patients compared to healthy individuals. tRF-25, tRF-38 and tRF-18 derived from plasma exosomes showed particular promise as their relative expression in osteoporosis patients, as validated and confirmed by qRT-PCR, was significantly higher than in healthy controls. Importantly, the tRF panel made up of these three tRFs underwent receiver operating characteristic (ROC) curve analysis and showed a remarkable average area under the curve (AUC) of 0.815, which is higher than other known miRNA biomarkers used for osteoporosis. Another degenerative bone disease where exosomal tRNAs showed significant changes in levels was identified in patients suffering from steroid-induced osteonecrosis of the femoral head (SONFH). Fang et al. [61] compared the expression profile of exosomal tsRNA in a small number (*n* = 10) of SONFH patients with healthy controls, and further sought to characterise the effect bone marrow-derived stem cell (BMSC) exosomes had on the pathogenesis of SONFH. The authors found three exosomal tsRNAs expressed at significantly lower levels in SONFH patients, validated by qRT-PCR. Importantly, their target genes were all implicated in the Wnt-signalling pathway and osteogenic differentiation. The authors further focussed on tsRNA-10277, as it had the largest fold change and found tsRNA-10277 containing BMSC exosomes had beneficial effects by decreasing adipogenesis in SONFH cells and promoting osteogenesis, directly contrasting the effect dexamethasone had on BMSC exosomes in SONFH patients. Confusingly, the authors have used an idiosyncratic nomenclature which does not correspond to any known tRF.

Mesenchymal stem cells (MSCs) are a common source of EVs for study due to their ability to be cultured easily as well as multi-lineage differentiation, giving rise to a variety of mesenchymal phenotypes such as osteoblasts (bone), adipocytes (fat), and chondrocytes (cartilage). A study by Baglio et al. [62] compared a number of primary BMSCs and adipose stem cells (ASCs) from elective plastic surgical samples to assess the EV RNA profiles released during culture. Sequencing analysis showed that whilst ASC exosomes showed a major RNA profile corresponding to tRNA-halves, BMSC exosomes showed two distinct RNA profiles. Further investigation into the genes involved in pluripotency revealed that the two RNA profiles observed in BMSC exosomes were each related to the different BMSC differentiation stages, with more differentiated BMSCs associated with a predominant tRNA-half profile (much like ASCs), and less differentiated BMSCs correlating with a profile of full-length tRNAs and 33-nt fragments. In addition, by comparing cells and their corresponding EVs, they were able to highlight a weak correlation and therefore selective incorporation of RNA species into EVs. EV tRNAs accounted for 5–11% of RNA content in cells but represented >50% of total small RNAs in ASC EVs and 23–35% in BMSC EVs. The high abundance of tRNAs found in the Evs thus prompted further investigation into their subtypes, where 5′-tRNA halves were found to be the predominant species. EVs in BMSCs were also explored by Yan et al. [63] as part of a study into osteoblastogenesis, using UC to first isolate the EVs. Subsequent RNA sequencing was used to identify the miRNA and tRNA profiles from the Evs derived from ASCs and BMSCs, which were then compared with parental cells prior to differentiation. Similar results to the study by Baglio et al. were found, where although tRNA reads in the BMSC EVs decreased from 52% to 14%, from Day 0 to Day 7 of osteoblastic differentiation, this was not noted in ASC EVs. The striking fall in tRNA expression levels was found to be mainly contributed by a decrease in 5′-tRNA halves (Gly-CCC, Gly-GCC, and His-GTG). In addition, three tRNA fragments, which were all 5′-tRNA halves, constituted over half the tRNA reads in each cohort, indicating a degree of heterogeneity. Although this was a small study, it appeared to show that alterations in this pattern of expression were seen during cell-specific osteoblastic differentiation. Another study conducted by Kaur et al. [64] aimed to characterise the RNA profile of the EVs from adipose tissue mesenchymal stem cells (AT-MSCs) and human pluripotent stem cells. Interestingly, the isolation of EVs was performed via a combination of UC and miRCURY Exosome Isolation Kit (Exiqon now QIAGEN). The authors subsequently found that whilst most EV reads were attributed to rRNAs, most of the sRNA reads in hPSCs and AT-MSCs were still nevertheless attributed to tRNAs at 69% and 47%, respectively. The selectivity of AT-MSCs for tRNAs shown here is again in line with the findings of Baglio et al.

#### 2.4.6. EV tRNAs in Infectious Disease

Human immunodeficiency virus (HIV) infection, tuberculosis, and malaria rank consistently among the top few infectious diseases which cause the greatest burden of disease worldwide. Interestingly, three of the four studies identified here were on these three diseases.

Chettimada et al. [65] investigated and compared the plasma EV RNA profile of 20 HIV-positive patients with 20 HIV-negative controls. Notably, the cohort of HIV-positive patients was on antiretroviral therapy and had viral loads <1000 HIV RNA copies/mL. They noted that the number of unique mapped reads corresponding to tRNAs were equal between HIV-positive patients and uninfected individuals but did not comment on the differential expression of particular tRNAs. Another infectious disease closely associated with HIV infection is tuberculosis. Pawar et al. [66] investigated the role of 5′ tRNA halves during infection of human monocyte-derived macrophages (HMDMs) using Mycobacterium bovis (BCG). EVs were isolated via ultracentrifugation and tRNAs were validated via qRT-PCR. The authors found that 5′-tRNA-His-GUG halves were in particular selectively packaged into EVs of HMDMs and present to a high degree, approximately 210-fold higher than the most abundant miRNA (miR-150) found. 5′-tRNA-His-GUG halves were found to stimulate Toll-like receptor 7 (TLR7) pathways significantly but were also highly specific for TLR7. In addition, 5′-tRNA-His-GUG halves and 5′-tRNA-Glu-CUC halves were found to be upregulated by TLR-activated NF-κB, where increased expression of ANG mRNA led to the increased cleavage of tRNAs. Furthermore, a feed-forward loop was suggested where the tRNA-halves could potentially activate TLR7 pathways, also leading to NF-κB activation and, thus, further tRNA-half production. Importantly, as NF-κB is linked to many inflammatory diseases and malignant conditions; this could, in turn, suggest the involvement of tRNA-halves found within EVs in such diseases. Malaria is another disease which is challenging to treat and requires long term management. Sisquella et al. [67] investigated the RNA profile from the EVs secreted by P. falciparum-infected red blood cells to determine whether EVs contained parasitic small RNA. They found a significant increase in RNA content after infection and distinct tRNA peaks when quantified by RNA Bioanalyser.

In addition, a study by He et al. [68] investigated the microbial–host relation in the oral cavity by exploring the tRNAs contained within EVs in the saliva and their effects on the growth of oral bacteria, particularly those that had high sequence similarity to tRNAs from a group of Gram-negative oral bacteria. The authors found that these 5′-tRNA halves (tsRNA-000794 and tsRNA-020498) in salivary exosomes inhibited growth and protein synthesis of Fusobacterium nucleatum, a key oral commensal and opportunistic pathogen. These EV tRNA fragments in turn were increased in salivary EVs in response to F. nucleatum. This study, therefore, suggests a possible mechanism evolved by the oral microbiota to suppress the growth of Gram-negative organisms to maintain oral health through microbial–host EV-mediated homeostasis. 

#### 2.4.7. EV tRNAs in Diseases of the Reproductive Tract 

Two studies have been identified in this review where tRNAs in EVs play potential roles in the diseases of the reproductive tract. Zhang et al. [69] reported the use of exosomal tRFs to predict whether microdissection testicular sperm extraction (Micro-TESE) could be used to predict success as a sperm retrieval procedure for patients suffering from non-obstructive azoospermia (NOA). Differential expression of EV tRFs was analysed from the blood plasma of twelve patients with successful micro-TESE against eighteen patients where no sperm was retrieved. This initial screen for differentially expressed tRFs yielded three potential candidates. However, after qPCR validation, only two tRFs (tRF-Gly-GCC-002 and tRF-Glu-CTC-005) were found to reach statistical significance. ROC curve analysis was performed and revealed that tRF-Gly-GCC-002 and tRF-Glu-CTC-005 could predict NOA with successful sperm retrieval with an AUC of 0.921 and 0.954, respectively, indicating high predictive accuracy. The other study grouped into this category was an investigation into the role of human follicular fluid EVs in polycystic ovarian syndrome (PCOS) by Hu et al. [70]. Four PCOS patients and two healthy patients were included in the study. With a fold change >1, ten tRNAs were found upregulated in PCOS patients whilst another ten tRNAs were significantly downregulated. The authors, however, did not further investigate these findings as only miRNAs were the focus of the study. 

#### 2.4.8. EV tRNAs in Autoimmune Disease

Four studies reviewed EVs in autoimmune diseases including systemic lupus erythematosus (SLE), atopic dermatitis in the young, and dermatomyositis. Yang et al. [71] was a biomarker study performed on a very large sample of patients including 192 SLE patients and 109 normal controls to examine the difference in serum tsRNA profile between SLE patients with and without lupus nephritis and healthy controls. They identified tRF-His-GTG-1 as the most significantly upregulated in the serum, between SLE patients without lupus nephritis and control patients. They then conducted a separate experiment on an unspecified number of samples to show that this marker tRF-His-GTG-1 was also present in exosomes (*p* < 0.0001). Dou et al. [72] revealed the potential role of exosomal tRNAs as a way of inhibiting the pro-inflammatory process mediated by macrophages commonly seen in SLE. In order to investigate the effect MSC exosomes had on macrophages, the authors compared the tRNA expression profile of macrophages cultured with MSC exosomes with macrophages which were not. They found tsRNA-21109 as one of the prominent tRNAs differentially upregulated. They then investigated macrophage activation, using MSC exosomes which either contained or did not contain tsRNA-21109, and found MSC exosomes containing tsRNA-21109 inhibited the pro-inflammatory activation of macrophages and thus could represent a potential target for SLE treatment.

Meng et al. [73] for the first time attempted to elucidate the role of exosomal tRFs in paediatric patients with atopic dermatitis. Using plasma samples from three atopic dermatitis patients and three healthy controls, the authors revealed an impressive 135 exosomal tRFs which are differentially expressed. Of these, two select tRFs were chosen to be validated via qRT-PCR in 20 patients and 20 controls showing significant downregulation in patients compared to controls. These two tRFs then underwent ROC curve analysis and, in particular, tRF-28-QSZ34KRQ590K showed a remarkable AUC of 0.900 for diagnosing atopic dermatitis. However, tRF-28-QSZ34KRQ590K did not show any significant correlation with the disease severity of atopic dermatitis. Another autoimmune condition was examined by Zhong et al. [74], where the exosomal RNA profile from the blood samples of 10 dermatomyositis patients, with half of the patients having simultaneous interstitial lung disease (ILD) involvement, was investigated. In all three patient groups, namely dermatomyositis patients with ILD and without ILD and control patients, exosomal tRNA represented one of the smallest proportions of sRNA, with 0.42%, 0.49% and 0.64%, respectively. 

#### 2.4.9. EV tRNAs in Other Diseases

Two studies finally highlight the diversity of EV tRNA in other diseases. Lee et al. [75] found that in patients suffering from Post-Traumatic Stress Disorder (PTSD), all patient samples contained EVs with significantly higher proportions (*p* < 1 × 10^−4^) of tRNAs compared to EV depleted plasma and plasma. Fitz et al. [76] investigated the potential of using plasma EV associated RNAs to differentiate between Alzheimer’s disease (AD) patients and normal controls who have no dementia. tRNAs were found to contribute only 4.7% of the genes mapped. Small nucleolar RNAs on the other hand were found to have significant discriminatory power for identifying AD patients.

## 3. Discussion

In our review, we have focussed on the tRNAs, tRNA-halves, and tRFs associated with human EVs, in health and pathological disease states. We have described the various biofluids used for EV capture and characterised the different EV isolation methods used in protocols for the subsequent identification of EV tRNAs contained within. Importantly, we have shown that multiple pathologies and especially malignancies can be distinguished by differential EV tRNA expression and may thus be of clinical significance for both diagnosis and prognosis of the disease.

The abundance of tRNAs contained within EVs is highly variable and may depend on tissue and disease type, ranging from as low as 0.04% in the blood plasma EVs of gastric cancer patients [41] to as high as >95% from EVs obtained from healthy placental syncytiotrophoblasts [32]. However, tRNAs appear to be a consistent finding in EVs, occurring in sequencing samples with as high a prevalence (and sometimes higher) as miRNAs. Additionally, EV isolation methods will need to be tailored to the requirements of the experiment, with highly selective methods such as density gradient preferable when contamination with RBP is to be avoided. We found that 46.5% of the protocols used precipitation as their EV isolation method, and this has been recognised as having relatively low purity [77,78]. Moreover, the technological limitations of sequencing methods may further add to the complexity of interpreting the EV tRNAs found. Our review highlighted a few limitations of current sequencing technologies. In many studies, the library preparation is limited to specific size ranges [40] and low abundance species can lead to selective biases in enrichment. [21] Moreover, conventional sequencing methods may not effectively capture RNAs that are post-transcriptionally modified [23]. Finally, Tosar et al. [79] highlighted how piRNA transcripts differ from some tRNA-halves by only one nucleotide. Individual piRNA and as well as other small cytoplasmic RNA species are highly homologous to major tRNA fragments within commonly used databases; thus, mapping of reads may lead to an under- or over-representation of tRNA fragments. The limitations encountered with sequencing technologies could be addressed through novel pipelines such as that reported by Amorim et al. [22]. Ongoing collaborative work such as the Extracellular RNA Communication Consortium has made efforts to improve this and work towards strategies such as deconvolution to determine tRNA sources [80].

It is increasingly recognised that tRNA-derived fragments have a role in mammalian cells, with 5′-tRNA-halves having been associated with actions on the ribosome leading to stress granule formation [81] and translational inhibition. Smaller tRFs share a size similarity to miRNAs and there is significant evidence in the wider (non-EV) literature that they can have actions in conjunction with Argonaute [82], the Piwi subfamily, and other RNA-binding proteins [82,83].

tRNAs and their fragmentation products have been found as an EV biomarker in circulating fluid and tissue for a number of malignancies including leukaemia, prostate, ovarian, pancreatic, and colorectal cancers [84,85]. In our study, breast cancer showed particular promise with several differentially expressed tRNAs (tRF-Arg-CCT-017, tRF-Gly-CCC-001, and tiRNA-Phe-GAA-003) associated with key regulatory pathways such as Wnt signalling, classically involved in cancer initiation and progression. [86] In fact, several other malignancies such as colorectal cancers [87], gastric cancers [88], and glioblastomas [89] also secrete cancer-stem-cell-associated EVs, which act on the β-catenin/Wnt-signalling pathways to increase stemness, in turn increasing their tumorigenic potential. 

In addition, the majority of the studies in our review showed that the fragmentation products of tRNAs, namely 5′-tRFs and 5′ tRNA-halves, were the most abundant sub-species of tRNAs identified in EVs. A possible reason for the observation of 5′ products was shown to be potentially due to reverse transcription failure due to post-transcriptional modifications in otherwise full-length tRNAs by Shurtleff et al. who used an alternative thermostable sequencing system. What is also clear is that simple analysis of tRNA molecules by broad amino acid (isoacceptor types) may not be sophisticated enough to determine potentially clinically relevant changes in EV tRNA profiles. Further validation of sequencing by qRT-PCR is necessary to prove true differential expression and this means additionally incorporating the standardised use of tRNA molecular terminology.

We have also highlighted that several biological mechanisms have been clearly associated with changes in tRNA-halves levels. For instance, osteoblastogenesis revealed that BMSCs alter their cargo content in EVs from full-length tRNAs to tRNA-halves as their differentiation progresses [62]. We have noted the evidence that there are time-dependent changes in the EV tRNA profile of cancer patients and may perhaps reflect clinical changes such as disease burden [36]. In addition, tRFs have even been shown to possess potential diagnostic potential in a multitude of diseases.

tRNA-sorting mechanisms in EVs have also been the focus of various studies, with RNA-binding proteins [23], surface markers [26], and tRNA enzymatic stability as some of the mechanisms identified [25]. Potential therapeutic uses of EV tRNA include the loading of synthetic oligonucleotides into EVs via parent cells to alter downstream gene expression in donor cells [25]. This particular paper by Gambaro et al. [25] demonstrated key mechanisms for tRNA sorting, such as the concentration-dependent transfection and selection of stable synthetic oligonucleotides against enzyme degradation, through well-controlled transfection experiments.

Due to the heterogeneity of the studies included in our review, a meta-analysis could not be conducted; thus, a narrative synthesis of the literature was undertaken. Many studies may include tRNA data but not explicitly mention this and will have been excluded. We also restricted our search to articles involving human cell models or patient-derived biofluids which may exclude important studies using animal EV models that could have shed some light on tRNA markers across other species.

### Further Work and Outlook

Our review shows that EV tRNAs are present in various healthy and diseased cells and tissues and confirms that tRNAs and their fragmentation products are one of the most abundant RNA species expressed in EVs and cells after miRNAs. Coupled with their presence in diverse biofluids, we believe this represents a clinical significance from an untapped source of potential markers. There is a vital need for future studies to determine the origin and loading of EV tRNAs, as well as the proteins or nucleases involved in their fragmentation and modification. In the future, we see combinations of tRNAs with other RNA species (such as miRNA) found from EVs will be more likely to yield better diagnostic and prognostic accuracy. Their differential expression presents an opportunity for further clinical research where EV-associated tRFs are understood as a mechanism of cellular stress and signalling. It is in this context that future protocols must include size selection, as the question of whether EV tRNAs exist as fragmentation products or mature tRNAs is yet to be determined. Finally, collaborative work through online repositories of EV RNA sequencing data will ensure that future biomarkers remain available for data scientists and bioinformaticians. Some databases to manage and discover tRFs, such as trfdb (http://genome.bioch.virginia.edu/trfdb), tRFun (https://rna.sysu.edu.cn/tsRFun/), tRFexplorer (https://trfexplorer.cloud/), and MINTbase v2.0 (http://cm.jefferson.edu/MINTbase/) have arisen to promote this effort.

## 4. Materials and Methods

### 4.1. Objectives

Our primary aim was to perform a systematic review to identify all studies involving tRNA cargo present associated with human-derived EVs isolated in vivo or in vitro. Our objectives included defining the EV source, EV characteristics, and results of these studies in order to determine the most up-to-date knowledge on the functional and clinical significance of EV tRNA cargo.

### 4.2. Search Strategy

Electronic bibliographic databases MEDLINE (Medical Literature Analysis and Retrieval System Online) and EMBASE (Excerpta Medica Database) were searched according to a systematic pre-specified search strategy which included terms such as “extracellular vesicles”, “microvesicles”, “exosomes”, “transfer ribonucleic acid”, and “tRNA fragments” (see Appendix B). An iterative search involving bibliographies of the included articles was also manually searched to identify any additional relevant articles not found using the initial search strategy. The last search was conducted on 1 January 2022. 

### 4.3. Screening and Selection Criteria 

All articles identified from the search strategy were uploaded onto an online collaborative tool Covidence (Melbourne, Australia) and de-duplication was undertaken automatically. Title and abstract screening and full-text screening was conducted by (Q.Z.C.Y) and (D.L) independently and cross-referenced for accuracy. Any disagreements were reviewed and resolved after discussion. The screening process is shown in the PRISMA flowchart (Figure 3).

Papers that were included for data extraction included original research articles which investigated RNA sequencing data from EVs derived from human biofluids, human pathogens, or human cell lines, either healthy or pathological. Conference abstracts, poster presentations, review papers, and research articles that solely investigated EVs of non-human origin were excluded as well as bioinformatics papers that drew on prior datasets characterising or identifying tRNAs in EVs.

## Figures and Tables

**Figure 1 ijms-23-03692-f001:**
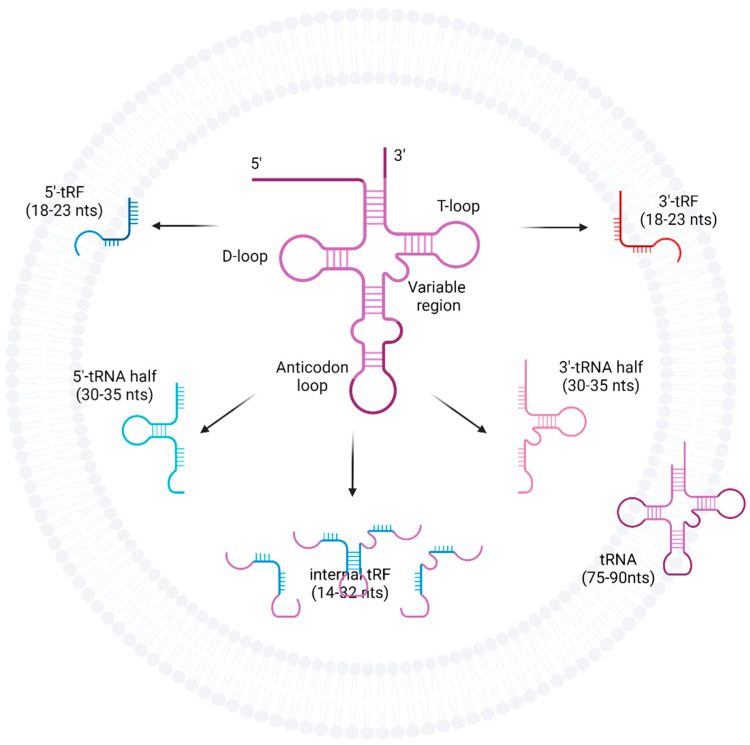
Fragmentation of tRNAs into tRFs, tRNA-halves, and internal tRFs. tRNAs are cleaved at the anti-codon loop by enzymes such as angiogenin, Dicer, or RNase Z into either 5′ or 3′-tRNA-halves. tRFs can arise from mature tRNA, pre-tRNA as well as tRNA-halves, and are formed when cleavage occurs at either the D-loop or T-loop. However, the enzymes responsible for tRF production are less well understood.

**Figure 2 ijms-23-03692-f002:**
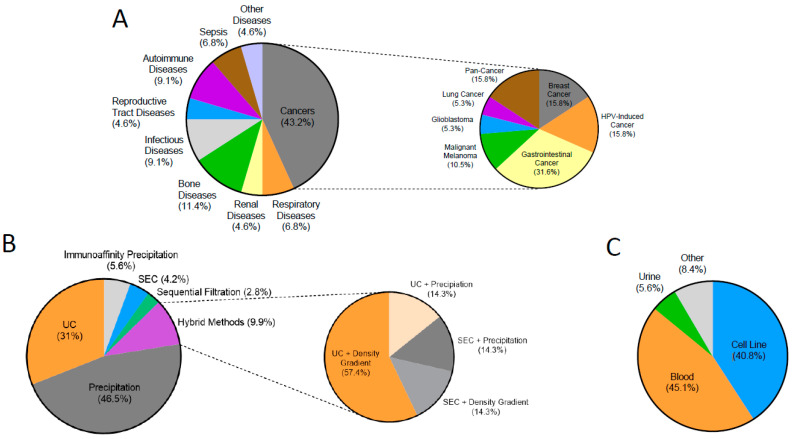
Types of diseases included in the review and the EV methodologies used. (**A**) Pie chart showing the primary disease focus of the studies (n = 44) included in the review. Studies which investigated EVs in healthy biofluids or methodology optimisation were not included in the pie chart; (**B**) Pie chart showing the different EV isolation methodologies used in the protocols investigated (n = 71); (**C**) Pie chart showing the different types of biofluids used for EV isolation in the protocols investigated (n = 71).

**Figure 3 ijms-23-03692-f003:**
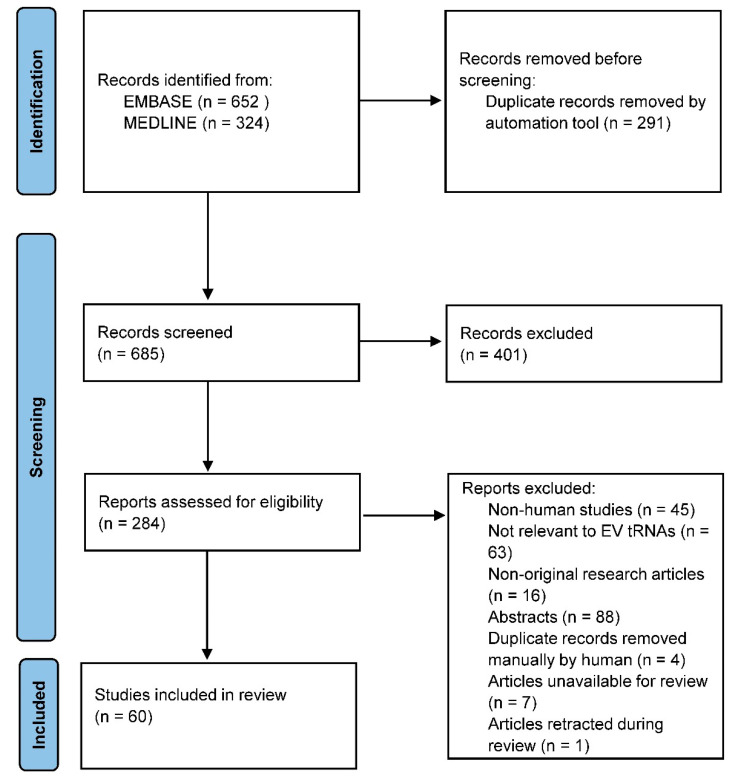
PRISMA flowchart showing the screening process of included articles.

## Data Availability

Not applicable.

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
