# Peer review of "The Clinical Significance of Transfer RNAs Present in Extracellular Vesicles"

_ijms, 2022, doi:10.3390/ijms23073692_

Round 1

Reviewer 1 Report

Despite considerable interest in the RNA cargo of extracellular vesicles (EVs), relatively little attention has been paid to their content in transfer RNA, which helps decoding an mRNA sequence into a protein. The review of Liu et al helps filling this gap, providing  a wealth of information that will be of interest both for researchers in this specific field and to improve our understanding of the complex EV biology. I have no particular observation: nice piece of work.

Author Response

Dear Reviewer 1. Thank you for your time and kind reply. We agree that this should hopefully be a timely review of the literature regarding EV tRNA expression in health and disease.

Reviewer 2 Report

The paper is complete and clear, cover the topic. I do not have corrections stricto sensu on the paper (maybe just clarify extravesicular and extracellular in some parts)

However, I feel it lacks a few discussions about : 

1) the putative effect/MOA of tRNA on recipient cells

2) explanation of why tRNA are particularly interesting compared to other (small) RNAs

3) in the end => a critical discussion on clinically relevant opportunities

Best regards

Author Response

Dear Reviewer 2,

Thank you for your time and pertinent questions regarding our review. Please find a response below.

With regards point 1: “the putative effect/MOA of tRNA on recipient cells”

Thank you for your recommendation to add a point on the effect on cells. This is a wide-ranging question which is yet to be fully elucidated, has been demonstrated on all aspects of the post-transcriptional pathway, as well as being cell-specific, condition-dependent and would warrant a full discussion on its own. We have therefore aimed to sum this up with the paragraph below which I hope will direct the interested reader to further relevant reviews. This has been inserted after paragraph 3 of Discussion section.

“It is increasingly recognised that tRNA-derived fragments have a role in mammalian cells, with 5’-tRNA-halves have been associated with actions on the ribosome leading to stress granule formation and translational inhibition. Smaller tRFs share a size similarity to miRNAs and there is significant evidence in the wider (non-EV) literature that they can have actions to degrade and cleave mRNA in conjunction with Argonaute, the Piwi subfamily and other RNA binding proteins.”

Regarding point 2 and 3, “explanation of why tRNA are particularly interesting compared to other (small) RNAs” and “clinically relevant opportunities”.

Thank you for your recommendations to add information on the general interest for other readers interested in small RNAs. We have decided to combine this with the question of clinically relevant opportunities, to explain the “state of the art” for researchers. It is clear that the increasing complexity of RNA biology has presented us with a dilemma regarding which small RNAs are likely to be pertinent in health and disease. We feel that tRNAs require recognition as highly represented in non-coding sequencing data and therefore are under investigated as a whole. This will naturally lead to a clear clinically relevant opportunity, in two ways, firstly as a potential signature and mechanism of cellular stress in future experiments, but also as a hitherto unevaluated signature within publicly available RNA-seq collaborative datasets. The paragraph edited and included here is within our conclusion and outlook:

“Our review shows that EV tRNAs are present in various healthy and diseased cells and tissues and confirms that tRNAs and its fragmentation products are one of the most abundant RNA species expressed in EVs and cells after miRNAs. Coupled with their presence in diverse biofluids, we believe this represents a clinical significance from an untapped source of potential markers. There is a vital need for future studies to deter-mine the origin and loading of EV tRNAs, as well as the proteins or nucleases involved in their fragmentation and modification. In future we see combinations of tRNAs with other RNAs species (such as miRNA) found from EVs will be more likely to yield better diagnostic and prognostic accuracy. Their differential expression presents opportunity for further clinical research where EV associated tRFs are understood as a mechanism of cellular stress and signalling. It is in this context that future protocols must include size selection, as the question of whether EV tRNAs exist as fragmentation products or mature tRNAs is yet to be determined. Finally collaborative work through online repositories of EV RNA sequencing data will ensure that future biomarkers remain available for data scientists and bioinformaticians.”

Kind regards.

Reviewer 3 Report

The current manuscript is well-written and ready for the publication.

Author Response

Dear Reviewer 3. Thank you for your time and rapid response. We do hope this review will be of interest to the reader involved in EV research.